# Biotin Is Required for the Zinc Homeostasis in the Skin

**DOI:** 10.3390/nu11040919

**Published:** 2019-04-24

**Authors:** Youichi Ogawa, Manao Kinoshita, Takuya Sato, Shinji Shimada, Tatsuyoshi Kawamura

**Affiliations:** Department of Dermatology, Faculty of Medicine, University of Yamanashi, Yamanashi 409-3898, Japan; mkinoshita@yamanashi.ac.jp (M.K.); d17sm039@yamanashi.ac.jp (T.S.); sshimada@yamanashi.ac.jp (S.S.); tkawa@yamanashi.ac.jp (T.K.)

**Keywords:** biotin deficiency, zinc deficiency, acrodermatitis enteropathica, Langerhans cells, adenosine triphosphate

## Abstract

Patients with biotin deficiency present symptoms that are similar to those in patients with acrodermatitis enteropathica (inherent zinc deficiency). However, the association between biotin and zinc deficiency remains unknown. We have previously shown that epidermal keratinocytes of mice fed zinc-deficient (ZD) diets secreted more adenosine triphosphate (ATP) than those of mice fed zinc-adequate (ZA) diets and that epidermal Langerhans cells are absent in ZD mice. Langerhans cells highly express CD39, which potently hydrolyzes ATP into adenosine monophosphate (AMP). Thus, a lack of Langerhans cells in ZD mice leads to non-hydrolysis of ATP, thereby leading to the development of ATP-mediated irritant contact dermatitis. In this study, we examined if biotin-deficient (BD) mice showed the same underlying mechanisms as those in ZD mice. BD mice showed reduced serum zinc levels, disappearance of epidermal Langerhans cells, and enhanced ATP production in the skin. Consequently, irritant contact dermatitis was significantly enhanced and prolonged in BD mice. In conclusion, the findings of our study showed that biotin deficiency leads to zinc deficiency because of which patients with biotin deficiency show similar symptoms as those with acrodermatitis enteropathica.

## 1. Introduction

Biotin is a water-soluble vitamin that serves as a co-enzyme for five carboxylases, namely, the covalently bound coenzyme for acetyl-CoA carboxylases 1 and 2, pyruvate carboxylase, propionyl-CoA carboxylase, and 3-methylcrotonyl-CoA carboxylase. These biotin-dependent carboxylases facilitate various metabolic reactions such as gluconeogenesis, fatty acid synthesis, and amino acid synthesis. Mammals cannot synthesize biotin by themselves; they obtain biotin through food and some gut microbiota produce biotin. Biotin deficiency (BnD) rarely occurs in people who consume a normal mixed diet [1,2]. However, genetic deficiency of holocarboxylase synthetase and biotinidase, continuous consumption of raw egg whites, parenteral nutrition, and modified milk without biotin supplementation could result in BnD [1,2,3]. Patients with BnD [4] are known to develop clinical symptoms that are similar to those in patients with acrodermatitis enteropathica (AE), which is caused by loss-of-function mutations in Zrt-, Irt-like protein (ZIP) 4 [5,6,7]. The clinical symptoms of AE include characteristic skin lesions, alopecia, and diarrhea.

Patients with AE exhibit severe zinc deficiency (ZnD) because their intestines lack the ability to absorb zinc. The characteristic skin lesions of AE occur in the periorificial, anogenital, and acral regions, which are exposed to the external environment. These lesions were caused by adenosine triphosphate (ATP)-mediated irritant contact dermatitis (ICD) [8,9,10]. Mice fed zinc-deficient (ZD) diet showed an impaired allergic contact dermatitis (ACD) in response to dinitrofluorobenzene (DNFB) compared to mice fed a zinc-adequate (ZA) diet. ZnD leads to impaired immune responses because of dysfunction in the immune cells. Thus, DNFB-mediated ACD is attenuated in ZD mice. However, ZD mice exhibited a significantly increased and prolonged ICD in response to croton oil (CrO) compared with ZA mice [8,9,10]. ATP is released from keratinocytes (KCs) in response to various environmental stimuli through lytic and non-lytic mechanisms, which results in ICD [11]. An ex vivo organ culture with CrO showed that the amount of ATP released from the skin of ZD mice was much greater than that released from the skin of ZA mice. Additionally, an injection of apyrase that hydrolyzes ATP into adenosine monophosphate (AMP) restored the increased and prolonged ICD caused by CrO application in ZD mice [8]. These results suggest that the prolonged ICD response in ZD mice was mediated via the excess ATP release by KCs in response to irritants.

Langerhans cells (LCs) are a subset of antigen-presenting cells that are distributed in the epidermis [12]. LCs but not KCs express CD39 (ecto-nucleoside triphosphate diphosphohydrolase 1; ENTPD-1) that potently hydrolyzes ATP into AMP [13,14,15], thereby assuming approximately 80% of the epidermal ATP hydrolysis [16]. Interestingly, epidermal LCs were absent in the skin lesions of patients with AE and of ZD mice [8,9,10]. The impaired ATP hydrolysis because of the disappearance of LCs leads to ATP-mediated inflammation in the epidermis, followed by the development of ICD. Therefore, the characteristic skin lesions in patients with AE are developed by aberrant ATP production from ZD–KCs and defective ATP hydrolysis due to loss of CD39-expressing LCs. 

Patients with BnD show similar characteristic skin lesions as patients with AE. Thus, we hypothesized that skin lesions in BnD are caused by the same underlying mechanisms as those in ZnD.

## 2. Materials and Methods

### 2.1. Study Approval

Murine studies were conducted with the approval of and in accordance with the guidelines for animal experiments of the University of Yamanashi. 

### 2.2. Animals and Diets

Five-week-old female BALB/c mice were purchased from Oriental Yeast Co. Ltd. (Tokyo, Japan). Mice were maintained under specific pathogen-free conditions throughout this study. Biotin-deficient (BD) and zinc-deficient (ZD) diets were purchased from CLEA Japan Inc (Tokyo, Japan). The mice were fed control (biotin-adequate; BA) or BD diet from 5 to 21 weeks of age. BA and BD diets were of almost the same nutritional quality, differing only in terms of biotin content. In another experiment (Figure 1B), the mice were fed ZD or control (zinc-adequate; ZA) diet from 5 to 11 weeks of age. 

### 2.3. Reagents and Antibodies

Dinitrofluorobenzene (DNFB), croton oil (CrO), and bis(2-acrylamidoethyl) disulfide (BAC) were purchased from Sigma-Aldrich (St. Louis, MO, USA). FITC-conjugated anti-mouse IA/IE and PE-conjugated anti-mouse CD45 mAbs were purchased from BioLegend (San Diego, CA, USA).

### 2.4. Quantification of Zinc Levels in the Serum

Serum zinc levels of BA, BD, ZA, and ZD mice were determined using the Zinc Quantification Kit (abcam, Cambridge, MA, USA) as per the manufacturer’s instructions.

### 2.5. Histological Examination

Skin specimens from the backs (Figure 1E) and ears (Figure 2E,F) of BA or BD mice were surgically removed, fixed in 4% paraformaldehyde overnight at 4 °C, and then dehydrated in 70% ethanol. Samples were embedded in paraffin. Sections were stained with hematoxylin and eosin.

### 2.6. Preparation of Epidermal Cell Suspensions

The dorsal back skin was removed and incubated with 0.5% solution of trypsin (type XI, Sigma-Aldrich, St. Louis, MO, USA) in PBS for 30 min at 37 °C to separate the epidermis from the underlying dermis. After the removal of the loosened dermis, the epidermal sheets were gently agitated with 0.05% DNase1 (Sigma-Aldrich, St. Louis, MO, USA) in PBS for 10 min, and the resulting epidermal cell suspension was passed through a nylon mesh to remove hair and stratum corneum prior to use.

### 2.7. Flow Cytometry

Single-cell suspensions (1 × 10^6^) of epidermal sheets from BA and BD mice (3 mice per group) were stained for LCs with FITC-conjugated anti–IA/IE and PE-conjugated anti-CD45 mAbs for 30 min at 4 °C. Live/dead discrimination was performed using propidium iodide (Sigma-Aldrich, St. Louis, MO, USA). After washing, samples were analyzed using a FACSCalibur flow cytometer (BD Biosciences, San Jose, CA, USA). 

### 2.8. Quantification of ATP Release from the Skin

Both ears from all the mice were taken immediately after sacrifice. Ear skins of the dorsal (back) side were used in the experiment. Ear skin explants were prepared by cutting into 8.0 mm circular pieces. The 2 pieces of ear skin were floated with the epidermis side upward in 12-well plates containing 4 mL PBS and incubated on ice for 10 min. Subsequently, 0.1% BAC was added to the culture. After 30 min, ATP concentrations in the supernatants were quantified using the luciferin-luciferase assay.

### 2.9. ACD and ICD Responses

For chemical induction of allergic contact dermatitis (ACD), all mice were topically treated with 20 μL of 0.5% DNFB dissolved in acetone/olive oil (4:1), which was painted onto the shaved abdomen at days 0 and 1. The ears were then challenged by application of 10 μL of 0.2% DNFB on the right ear and vehicle alone on the left ear on day 5. For chemically induced irritant contact dermatitis (ICD), all mice received topical application of 1% CrO on the right ear and vehicle alone on the left ear. Swelling responses were quantified (right ear thickness minus left ear thickness) by a third experimenter using a micrometer. 

### 2.10. Quantitative Real-Time PCR Analysis

Epidermal Cell Suspensions were prepared from back skins of BA and BD mice. To isolate keratinocytes (KCs), CD45^+^ cells were eliminated using MACS (Miltenyi Biotec, Bergisch Gladbach, Germany). Total RNA was extracted using QIAzol^®^ Lysis Reagent (Qiagen, Hilden, Germany) and RNeasy^®^ Plus Universal Mini kit (Qiagen, Hilden, Germany) as per the manufacturer’s instructions. Reverse transcription reaction was performed using ReverTra Ace^®^ qPCR RT Kit (Toyobo, Osaka, Japan) as per the manufacturer’s instructions. mRNA levels were determined using commercially available primer/probe sets (TaqMan^®^ Gene Expression Assay, Applied Biosystems, Foster City, Calif) and the AB7500 real-time PCR system (Applied Biosystems, Foster City, Calif). The amount of target gene mRNA obtained using real-time PCR was normalized against the amount of the housekeeping control gene (*β*-actin) mRNA. Primers corresponding to murine *TGF-β*, *Il-34*, *Bmp7, Itgαv*, *Itgβ6*, and *Itgβ8* were designed by Takara (Shiga, Japan).

### 2.11. Statistics

Significant differences between experimental groups were analyzed by Student’s t test. *p* values less than 0.05 were considered significant. The survival of BA and BD mice was analyzed in the Kaplan–Meier format using log-rank (Mantel–Cox) test (Figure 1D).

## 3. Results

### 3.1. Dietary Biotin Deficiency Leads to Zinc Deficiency

After a 12-week biotin-adequate (BA; control) or biotin-deficient (BD) diet, the body size of BD mice was smaller than that of BA mice. Although BD mice did not exhibit apparent alopecia, the hair distribution was apparently sparse (Figure 1A). We next compared serum zinc levels between dietary zinc-deficient (ZD) and BD mice. Mice started to die after 8 weeks of initiation of ZD diet [8]. Accordingly, we measured serum zinc levels of zinc-adequate (ZA; control) and ZD mice at 7 weeks after the initiation of the ZA or ZD diet. Serum zinc levels were significantly impaired in ZD mice compared with ZA mice at this time point (Figure 1B; left panel). We found that BD mice showed a quick reduction in the serum zinc levels after initiation of the BD diet. After 9 weeks of initiation of the BD diet, the serum zinc levels in BD mice were significantly reduced compared with those in BA mice (Figure 1B; right panel), and were almost comparable to those in ZD mice after 7 weeks of initiation of ZD diet (Figure 1B; left and right panels). These data suggest that dietary biotin deficiency (BnD) leads to zinc deficiency (ZnD). 

The body weight of BD mice was significantly lesser than that of BA mice after 5 weeks of initiation of BD diet (Figure 1C). Moreover, the survival rate in BD mice was also significantly reduced in BD mice compared with that in BA mice after 13 weeks of initiation of the BD diet (Figure 1D). Akin to ZD mice, BD mice showed strong atrophy of fat tissues and an arrest in the hair cycle (Figure 1E). These data suggest that dietary BnD-mediated ZnD affects skin homeostasis. 

### 3.2. Dietary Biotin Deficiency Leads to the Development of ATP-Mediated Irritant Contact Dermatitis

We found that epidermal LCs (CD45^+^IA/IE^+^ cells) almost disappeared in BD mice (Figure 2A,B), thereby resulting in increased ATP production from the ear skins in response to irritants (Figure 2C). ACD was significantly impaired and ICD was significantly enhanced and prolonged in BD mice (Figure 2D,E). There was a massive neutrophil infiltration in both dermis and epidermis of BD mice compared to that in BA mice (Figure 2F; left panel). Histological examination of ICD lesions in BD mice revealed parakeratosis and cytoplasmic pallor, sub-corneal vacuolization, and ballooning degeneration of KCs (Figure 2F; right panel). These signs are histological features of cutaneous AE lesions in humans [17], whereas no such degeneration of KCs was observed in the ICD lesions of BA mice (Figure 2F; right panel). Several proteins are involved in the maintenance and survival of epidermal LCs, including transforming growth factor (TGF)-β [18], interleukin (IL)-34 [19], bone morphogenetic protein (BMP)-7 [20], integrin (ITG) αvβ6 [21], and ITGαvβ8 [22]. We examined if BnD alters the mRNA expression of these molecules in KCs. The expression of TGF-β was significantly downregulated in BD–KCs (Figure 2G). However, the expression of IL-34 was significantly upregulated in BD–KCs (Figure 2G). Interestingly, the expression of ITGαvβ6, but not ITGαvβ8, was significantly downregulated in BD–KCs.

## 4. Discussion

BnD causes abnormalities in the fatty acid composition of skin, such as accumulation of odd-chain fatty acids and abnormal metabolism of long-chain polyunsaturated fatty acids [4,23,24]. In this study, we found that dietary BnD leads to ZnD. However, the decline in the serum zinc levels in BD mice was relatively slower than that in ZD mice [8]. Accordingly, although changes in terms of the body weight and survival rate were similar to those observed in ZD mice, the kinetics in BD mice was slower than that in ZD mice [8]. LCs were found to be almost absent in the epidermis of BD mice, thereby leading to enhanced ATP production in skin, and similar findings were observed for ZD mice. This aberrant ATP accumulation in the skin results in the recruitment of neutrophils [25] and development of characteristic skin lesions in patients with ZnD and BnD. Indeed, decreased serum zinc levels have been reported in some patients with BnD [26,27,28]. On the other hand, some studies have demonstrated normal serum zinc levels in patients with BnD [29,30,31]. We speculate that this discrepancy might be attributable for the duration of BnD. Reduced serum biotin and zinc levels have been recently reported in patients with male androgenetic alopecia, suggesting the close association between biotin and zinc [32]. We investigated how dietary BnD led to ZnD but the precise mechanisms could not be elucidated (data not shown). Further analysis is required for elucidating the underlying mechanisms. 

Three groups of molecules hydrolyze ATP, namely, ENTPDs, ectonucleotide pyrophosphatase/phosphodiesterases (ENPPs), and alkaline phosphatase (ALP) [33,34]. The latter two molecules are zinc-dependent molecules. Among ENTPDs and ENPPs, ENTPD-1 (CD39), -2, -3, and -8 and ENPP-1, -2, and -3 hydrolyze ATP [33]. LCs strongly express ENTPD-1 (CD39) and weakly express ENTPD-2 and ENPP-1, -2, and -3. KCs weakly express CD39, ENTPD-2 and -3, and ENPP-1 and -2 [16]. Neither LCs nor KCs express ENTPD-8 or ALP. Therefore, although LCs strongly express CD39, other ATP-hydrolyzing molecules are weakly expressed in both LCs and KCs. LCs occupy approximately 3% of the epidermis, whereas KCs occupy approximately 97%. Regardless of this numeric difference between LCs and KCs in the epidermis, LCs perform approximately 80% of epidermal ATP hydrolysis, whereas KCs perform the remaining 20% [16]. Additionally, ZnD impairs the activity of ENPPs [35]. KCs weakly express ENPP-1 and ENPP-2. This explains one underlying mechanism by which ZD–KCs and BD–KCs show increased ATP production.

Both ZnD and BnD result in disappearance of epidermal LCs. TGF-β knock out (KO) mice, Langerin-Cre TGF-β1^fl/fl^ mice, Langerin-Cre TGF-βRI^fl/fl^ mice, and Langerin-Cre TGF-βRII^fl/fl^ mice lack epidermal LCs [36], suggesting that although TGF-β is produced by both LCs and KCs, LC-derived autocrine and/or paracrine TGF-β is critical for LC development and survival. TGF-β is secreted as an inactive latency-associated peptide (LAP)-TGF-β. LAP-TGF-β is processed by integrin (ITG) αvβ6 and/or αvβ8 expressed on KCs but not on LCs, which convert LAP-TGF-β into active TGF-β [36,37]. Thus, ITGαvβ6 or αvβ8 KO mice show substantially reduced number of epidermal LCs [21,22]. In BD–KCs, the mRNA expression levels of TGF-β and ITGαvβ6 were significantly downregulated. As described above, LC-derived, but not KC-derived, TGF-β is critical for LC homeostasis. In this respect, it is unclear how much downregulated TGF-β mRNA expression in BD–KCs is involved in disappearance of epidermal LCs. On the other hand, LAP-TGF-β produced by LCs is not processed into active TGF-β, because of downregulated ITGαvβ6 expression in BD–KCs. This may be one of underlying mechanisms for the loss of epidermal LCs in BD mice.

## 5. Conclusions

In this study, we found that dietary BnD led to ZnD. BD mice lost epidermal LCs possibly due to impaired ITGαvβ6 expression in KCs, as seen in ZD mice. This resulted in ATP accumulation in the epidermis, thereby leading to the development of ATP-mediated ICD lesions in the skin. We concluded that the characteristic skin lesions in patients with AE and BnD have common underlying mechanisms, and biotin is required for zinc homeostasis in the skin. 

## Figures and Tables

**Figure 1 nutrients-11-00919-f001:**
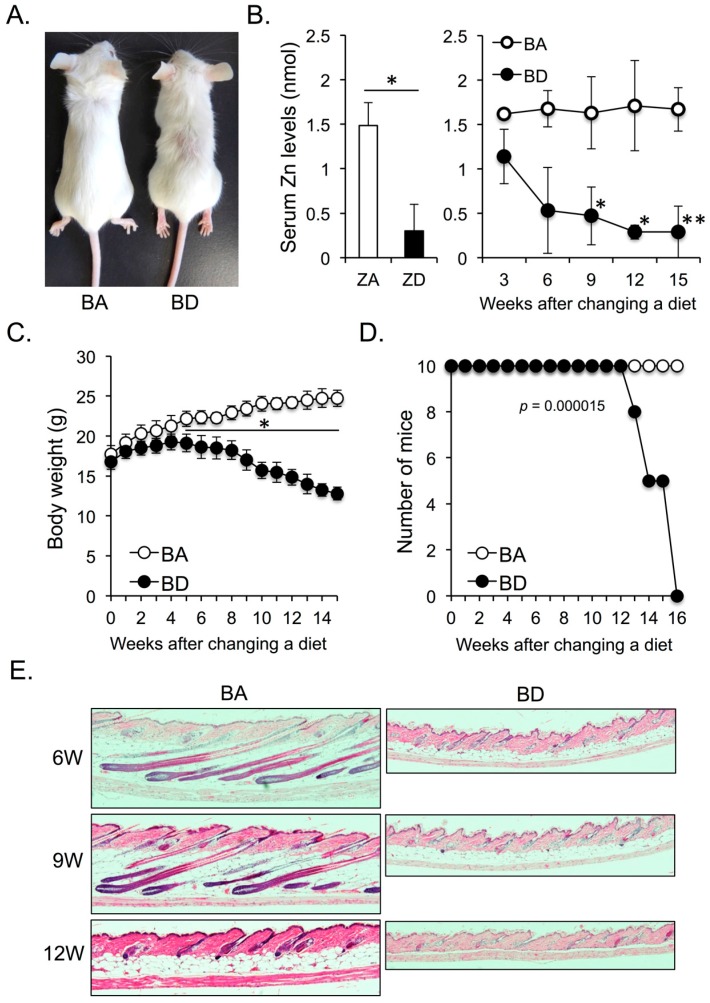
(**A**) Five-week-old female BALB/c mice were fed biotin-adequate (BA) (control diet) or biotin-deficient (BD) diet for 12 weeks. (**B**) Serum zinc levels of zinc-adequate (ZA) and zinc-deficient (ZD) mice after consumption of ZA (control) and ZD diet for seven weeks (left panel). Serum zinc levels of BA and BD mice fed BA (control) and BD diets for the indicated weeks (right panel). Five mice of each group were analyzed. Course of body weight (**C**) and survival rate (**D**) of BA and BD mice. Ten mice of each group were analyzed. (**E**) Change of skin phenotype of BA and BD mice after consumption of BA and BD diets for the indicated weeks (hematoxylin and eosin stain, ×100). Data are representative of three independent experiments. * *p* < 0.05, ** *p* < 0.01.

**Figure 2 nutrients-11-00919-f002:**
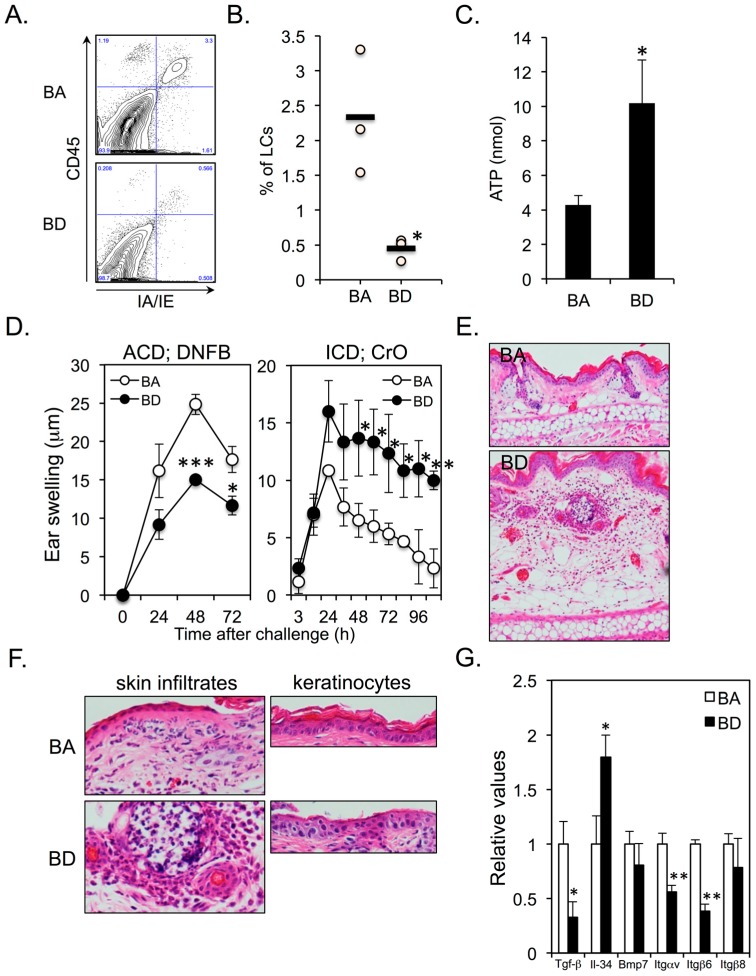
(**A**,**B**) Five-week-old female BALB/c mice were fed BA (control diet) or BD diet for 12 weeks. Epidermal langerhans cells (LCs) were identified as CD45^+^IA/IE^+^ cells. Three mice for each group were analyzed. (**C**) ATP release in response to BAC from the ear skins of BA and BD mice at 12 weeks after the initiation of BA and BD diets. Five mice of each group were analyzed. (**D**) ACD (left panel) and ICD (right panel) in BA and BD mice at 12 weeks after the initiation of BA and BD diets. Five mice of each group were analyzed. (**E**,**F**) ears of BA and BD mice elicited irritant contact dermatitis (ICD) response at 24 h after CrO application. (**G**) mRNA expression of molecules associated with LC differentiation and survival in KCs. Data are representative of 3 independent experiments. * *p* < 0.05, ** *p* < 0.01, *** *p* < 0.001.

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
