# Peer review of "Biotin Is Required for the Zinc Homeostasis in the Skin"

_nutrients, 2019, doi:10.3390/nu11040919_

Reviewer 1 Report

Dear Authors ,

Your article is very interesting and well-described. I have only one question about a sentence that does not seem clear to me:

line 185 you say : "we found that dietary BnD leads to ZnD ". I agree with this sentence.

but line 190 you say : "Although .... , ZnD might contribute to the developement of BnD when considering the similar symptom profiles." Are these two sentences not in contradiction? or can you explain me the significance ot the last sentence?

Best regards

Author Response

Your article is very interesting and well-described.

We deeply appreciate the positive comment.

1. I have only one question about a sentence that does not seem clear to me: 

line 185 you say: "we found that dietary BnD leads to ZnD ". I agree with this sentence. 

but line 190 you say : "Although .... , ZnD might contribute to the developement of BnD when considering the similar symptom profiles." Are these two sentences not in contradiction? or can you explain me the significance of the last sentence? 

We appreciate the critical suggestion. We deleted the sentence “ZnD might contribute to the developement of BnD when considering the similar symptom profiles”. Additionally, we made some corrections in the revised manuscript.

Reviewer 2 Report

The authors present a study of dietary biotin deficiency and find an association with reduced serum zinc and depletion of epidermal Langerhans cells. The results are of interest not only in the context of skin disorders but suggest a link between serum zinc and dietary biotin that as far as I am aware has not been previously described.

I think this last point could be emphasised and explored further in the discussion of the paper. For example, a recent paper (El-Esawy et al. Serum biotin and zinc in male androgenetic alopecia. J Cosmet Dermatol. 2019 Feb 3. doi: 10.1111/jocd.12865.) appears to provide support for such a link.
Other comments:

The results section is confusing to read due to inappropriate placement of discussion points. There is frequent comparison throughout the results with data from Zn deficient mice as reported by the same research group in reference 8. These comparisons are valid and of interest but should be made in the discussion rather than the results section. Similarly in Line 169 the sentence “ATP-mediated ICD was elicited…. Into the skin (18)” should be deleted as this is a discussion point and not a result presented in this paper. Deleting this sentence does not affect the message of the paragraph but would be of interest in the discussion. The results in general need to be looked at closely and discussion points taken out of the results and elaborated upon in the discussion instead.

Line 136. Symbols for alpha and beta missing in the integrin gene names.

Line 156. Saying the “decline in zinc was lower” is confusing. Change “lower” to “less”

Line 157 In this sentence “lesser” may be clearer than “lower”.

Author Response

The authors present a study of dietary biotin deficiency and find an association with reduced serum zinc and depletion of epidermal Langerhans cells. The results are of interest not only in the context of skin disorders but suggest a link between serum zinc and dietary biotin that as far as I am aware has not been previously described.

We deeply appreciate the proper summary and positive comment.

1. I think this last point could be emphasized and explored further in the discussion of the paper. For example, a recent paper (El-Esawy et al. Serum biotin and zinc in male androgenetic alopecia. J Cosmet Dermatol. 2019 Feb 3. doi: 10.1111/jocd.12865.) appears to provide support for such a link.

We appreciate the suggestion. We described about the suggested paper in the “Discussion” section of revised manuscript.

2. The results section is confusing to read due to inappropriate placement of discussion points. There is frequent comparison throughout the results with data from Zn deficient mice as reported by the same research group in reference 8. These comparisons are valid and of interest but should be made in the discussion rather than the results section. Similarly in Line 169 the sentence “ATP-mediated ICD was elicited…. Into the skin (18)” should be deleted as this is a discussion point and not a result presented in this paper. Deleting this sentence does not affect the message of the paragraph but would be of interest in the discussion. The results in general need to be looked at closely and discussion points taken out of the results and elaborated upon in the discussion instead.

We agreed with the reasonable suggestion. We deleted the descriptions about ZD mice from the “Result” section and moved to the “Discussion” section of revised manuscript.

3. Line 136. Symbols for alpha and beta missing in the integrin gene names.

We appreciate the suggestion. We corrected the suggested point.

4. Line 156. Saying the “decline in zinc was lower” is confusing. Change “lower” to “less”

We are sorry for the confused expression. We wanted to say that the decline in the serum zinc levels in BD mice was “relatively slower” than that in ZD mice, because the serum zinc levels in ZD mice reduced quickly and reached a plateau after two weeks of the initiation of ZD diet. Therefore, we changed “lower” to “relatively slower”.

5. Line 157 In this sentence “lesser” may be clearer than “lower”.

We appreciate the suggestion. We corrected the suggested point.

Round  2

Reviewer 2 Report

The authors have adequately addressed all points raised in the first review.